# Gene name errors: Lessons not learned

**Mandhri Abeysooriya , Megan Soria , Mary Sravya Kasu , Mark Ziemann** *

Deakin University, School of Life and Environmental Sciences, Geelong, Australia

* m.ziemann@deakin.edu.au

## Abstract

Erroneous conversion of gene names into other dates and other data types has been a frustration for computational biologists for years. We hypothesized that such errors in supplementary files might diminish after a report in 2016 highlighting the extent of the problem. To assess this, we performed a scan of supplementary files published in PubMed Central from 2014 to 2020. Overall, gene name errors continued to accumulate unabated in the period after 2016. An improved scanning software we developed identified gene name errors in 30.9% (3,436/11,117) of articles with supplementary Excel gene lists; a figure significantly higher than previously estimated. This is due to gene names being converted not just to dates and floating-point numbers, but also to internal date format (five-digit numbers). These findings further reinforce that spreadsheets are ill-suited to use with large genomic data.

**Data Availability Statement:** Code availability: Bash and R scripts supporting this paper are available in the GitHub repository (https://github.com/markziemann/GeneNameErrors2020).

**Funding:** The author(s) received no specific funding for this work.

## Author summary

Autocorrection is a feature of modern softwares including messaging apps, word processors and spreadsheets. These are designed to avoid data entry errors but "autocorrect fails" can lead to information being distorted in undesired and sometimes humorous ways. What is not funny though is having genomics spreadsheets suffer from auto-conversion of gene names like *SEPT8*, *DEC1* and *MARCH3* into dates, a problem first characterised in 2004. A 2016 article on this topic led the Human Gene Name Consortium to change many of these gene names to be less susceptible to autocorrect. Despite this, our work here shows that gene name autocorrect errors continue to accumulate in supplementary genomics spreadsheet files at a rapid pace. To avoid this and other reproducibility problems with spreadsheets, big changes are required in the way genomics scientists analyse and share data. We provide several practical steps researchers can take to avoid gene name errors and reiterate that big genomics data analysis is better suited to Python/R notebooks rather than spreadsheets.

## Background

It is a well-documented problem that spreadsheet software inadvertently converts gene symbols to dates and floating-point numbers, with these errors propagating downstream to annotation sets and other databases [1]. Previous work shows that gene name errors are made while

**Competing interests:** The authors have declared that no competing interests exist.

researchers analyse and prepare supplementary files for publication [2]. A screen of 18 journals found that one fifth of publications with supplementary Excel gene lists contained errors (704/3597). It remains unknown how frequent gene name errors are outside of these 18 journals, and whether the attention of previous publications has resulted in the mitigation of the problem.

Notably, software developers are beginning to remedy the problem at their end, with some packages like LibreOffice now resisting the conversion of gene symbols to dates (Version: 6.4.6.2). In addition, a recent announcement by the HUGO Gene Nomenclature Committee (HGNC) outlined plans for specifically changing gene symbols to avoid auto-correction [3]. For example, *SEPT1* becomes *SEPTIN1* and *MARCH1* becomes *MARCHF1*. It will likely take months and perhaps years for the new gene symbols to appear in publications.

Although changes to gene names and software will help, they won't solve the overarching problem with spreadsheets; that (i) errors occur silently, (ii) errors can be hidden amongst thousands of rows of data, and (iii) they are difficult to audit. Research shows that errors are surprisingly common in the business setting [4], which raises the question as to how common such errors are in science. The difficulty in auditing spreadsheets makes them generally incompatible with the principles of computational reproducibility [5].

Our main goal here is to examine whether gene name errors have diminished since 2016 or they continue to be a problem. We also assess the behaviour of current spreadsheet software in converting gene names to dates and identify Excel date genes across Eukarya. We follow this up with a screen of supplementary files from genomics-related PubMed Central (PMC) publications in the period 2014 to 2020.

# Results

## Testing spreadsheet software

We tested the propensity of various spreadsheet software to convert gene names into dates after importing a set of strings containing human gene names by (i) opening a text file, (ii) pasting data, and (iii) directly typing (Table 1). We found that Microsoft Excel and Google Sheets converted this data to dates in all three modes of import. LibreOffice and Gnumeric did not convert gene names to dates in our tests here. The date conversion behaviour of Excel and Google Sheets could be circumvented by formatting the destination cells as "plain text" prior to pasting or typing. Nevertheless, this result shows that using LibreOffice and Gnumeric are safer than Excel and Google Sheets.

## Identifying Excel date genes across kingdoms

Although recent changes have been made to human and mouse gene names to prevent conversion to dates [3], it is uncertain whether such changes have propagated through to other species. To assess this, we downloaded all eukaryotic gene names available in Ensembl and imported this into Excel and collected all genes that were converted to dates. In total there

**Table 1. Gene name conversion behaviour of spreadsheet software.**

| Software | Gene name conversion | | | |
| --- | --- | --- | --- | --- |
| | Microsoft Excel | Google Sheets | LibreOffice | Gnumeric |
| Text file open | Yes | Yes | No | No |
| Pasting data | Yes | Yes | No | No |
| Typing | Yes | Yes | No | No |

were 1,544 gene names converted to dates, from 104 taxa (Tables 2 and S1). Although most affected gene names were vertebrate in origin, there were gene names affected in all groups.

## Gene name errors by year

In order to determine whether gene name errors in supplementary files remain a problem, we undertook a screen of genomics-related publications in PMC. We collated a list of 166,139 genomics articles published between 2014 and 2020, and screened them using an enhanced script. In addition to identifying conversions to standard date formats (eg: 3/1/2016, Mar-3, 3-Mar) and floating-point numbers (eg: 9.33E+22), this script also recognises five-digit numbers as likely to be the result of gene name errors as this is the internal date format used by spreadsheets [1].

The results of this screen are shown in Table 3. From this set of publications, 32,841 had supplementary files in Excel format (with "xls" or "xlsx" suffixes). Of these, 11,117 publications were detected to contain at least one list of gene symbols. The software detected 3,470 publications with suspected gene name errors. After manually opening each spreadsheet file (5,136 files), we identified 34 publications as being false positives, leaving 3,436 publications with confirmed gene name errors (S2 and S3 Tables). These publications contain a total of 5,086 spreadsheets with gene name errors. The proportion of publications with Excel gene lists that contain errors was 30.9%; substantially higher than previously reported [2].

In the period 2014–2020, both the number of publications with Excel gene lists and the number of publications affected by gene name errors increased, with a pause in the period 2016–2018 (Fig 1A and 1B). On the other hand, the proportion of papers with Excel gene lists affected by errors remained stable over this period (Fig 1C). This result suggests gene name errors did not substantially reduce in the period after 2016 as we had hypothesized.

Next, to determine whether five-digit numbers explain the higher observed proportion of errors, we investigated a subset of 2160 affected spreadsheet files to determine frequency of error types. Dates in Mar-1 or 1-Mar format accounted for 1,797 files (83.2%). Errors in DD/MM/YYYY: format accounted for 19 files (0.88%) and 4 for floating-point numbers (0.18%). Five-digit numbers accounted for 340 files (15.7%) indicating that this error type is sufficiently common to account for the discrepancy between this and the previous report (S4 Table). When these five-digit numbers are formatted as standard dates, 292 (85.9%) appear in the months of March and September which is consistent with gene name errors.

## Gene name errors by organism

Next, we investigated whether the rate of gene name errors was dependent on the organism under study. We found that the frequency of gene name errors was highest for mouse and human datasets, while lower for Arabidopsis, chicken and rice (Table 4).

**Table 2. Gene names vulnerable to date conversion across Eukarya.**

|  | Taxa | Genes | Genes affected | Taxa affected |
|---|---|---|---|---|
| Vertebrates | 310 | 5,263,175 | 1,325 | 76 |
| Metazoa | 59 | 525,867 | 17 | 3 |
| Plants | 60 | 244,101 | 35 | 4 |
| Fungi | 59 | 788,221 | 140 | 12 |
| Protists | 39 | 163,026 | 27 | 9 |

**Table 3. Results of a screen for gene name errors in PMC.**

|  | 2014 | 2015 | 2016 | 2017 | 2018 | 2019 | 2020 | Total |
|---|---|---|---|---|---|---|---|---|
| Publications screened | 19976 | 21204 | 22261 | 23976 | 24986 | 26046 | 27690 | 166139 |
| Excel files screened | 2948 | 4318 | 4472 | 4355 | 4824 | 5481 | 6443 | 32841 |
| Excel files with gene lists | 2286 | 3037 | 3331 | 3021 | 3566 | 3342 | 4496 | 23670 |
| Publications with Excel gene lists | 936 | 1491 | 1579 | 1412 | 1653 | 1823 | 2223 | 11117 |
| Publications with suspected gene name errors | 284 | 490 | 477 | 443 | 475 | 594 | 707 | 3470 |
| False positive Excel files | 8 | 0 | 7 | 5 | 15 | 4 | 11 | 50 |
| False positive publications | 2 | 0 | 6 | 3 | 11 | 3 | 9 | 34 |
| Affected Excel files | 429 | 701 | 653 | 648 | 703 | 914 | 1038 | 5086 |
| Affected publications | 282 | 490 | 471 | 440 | 464 | 591 | 698 | 3436 |
| Proportion of publications affected (%) | 30.1% | 32.9% | 29.8% | 31.2% | 28.1% | 32.4% | 31.4% | 30.9% |

## Gene name errors by journal

In this sample of PMC articles 4,581 journals were represented. Of these, 741 journals published one or more supplementary Excel gene lists. There were 414 journals with at least one supplementary file with gene name errors. Next, we focused on journals that published at least 50 articles with supplementary Excel gene lists, finding 37 journals that accounted for 67.9% of affected publications (Table 5). The journals with the most affected articles included *Nature Communications*, *PLOS ONE*, *Scientific Reports*, *BMC Genomics*, *PLoS Genetics* and *Oncotarget*, with at least 100 affected publications each.

From the set of 37 journals, those with lowest rate of affected articles (<25%) included *Frontiers in Plant Science*, *BMC Plant Biology*, *G3*, *BMC Bioinformatics* and *PLoS Computational Biology*, while the journals with the highest rate (>40%) included *Cell*, *EBioMedicine*, *PNAS*, *Aging*, *Cell Reports*, *Nature* and *Oncogene*.

Next, we assessed whether there was any correlation between error proportion and the journal impact factor (JIF) for the set of 37 journals. A scatterplot of JIF and proportion of affected articles is shown in Fig 2. A correlation analysis indicated a statistically significant association using the Pearson (p = 0.0052, r = 0.462,) and Spearman (p = 1.95E-04; $\rho$ = 0.589) methods.

Next we assessed the temporal trends for the three journals with most gene name errors (Fig 3). *Nature Communications* showed a strong increase in articles with Excel gene lists and gene name errors over the period, while the proportion of affected articles recorded an increase from 33.3% to 39.5% in the period 2014–2020. *PLOS ONE* showed a trend of decreasing numbers of articles with supplementary Excel gene lists and number of affected articles but the proportion of affected articles was relatively flat over this time. *Scientific Reports* recorded a strong increase in articles with supplementary Excel files in the period 2014–2017 but has since remained stable. The proportion of affected articles in this journal did not show any consistent trend over this period.

## Novel error types

While we are familiar with common *SEPT* and *MARCH* conversions, we observed a variety of additional novel error modes. Some of these were likely related to locale language settings. In a few cases, the human gene *AGO2* was converted to Aug-02 (eg: PMC5537504 & PMC6244004), which may be due to Excel working in languages such as Italian, Spanish or Portugese. Similarly, the gene *MEI1* was seen to be converted to May-01 (eg: PMC6065148 & PMC5877863) and could be due to the similarity with the Dutch (mei). In one article

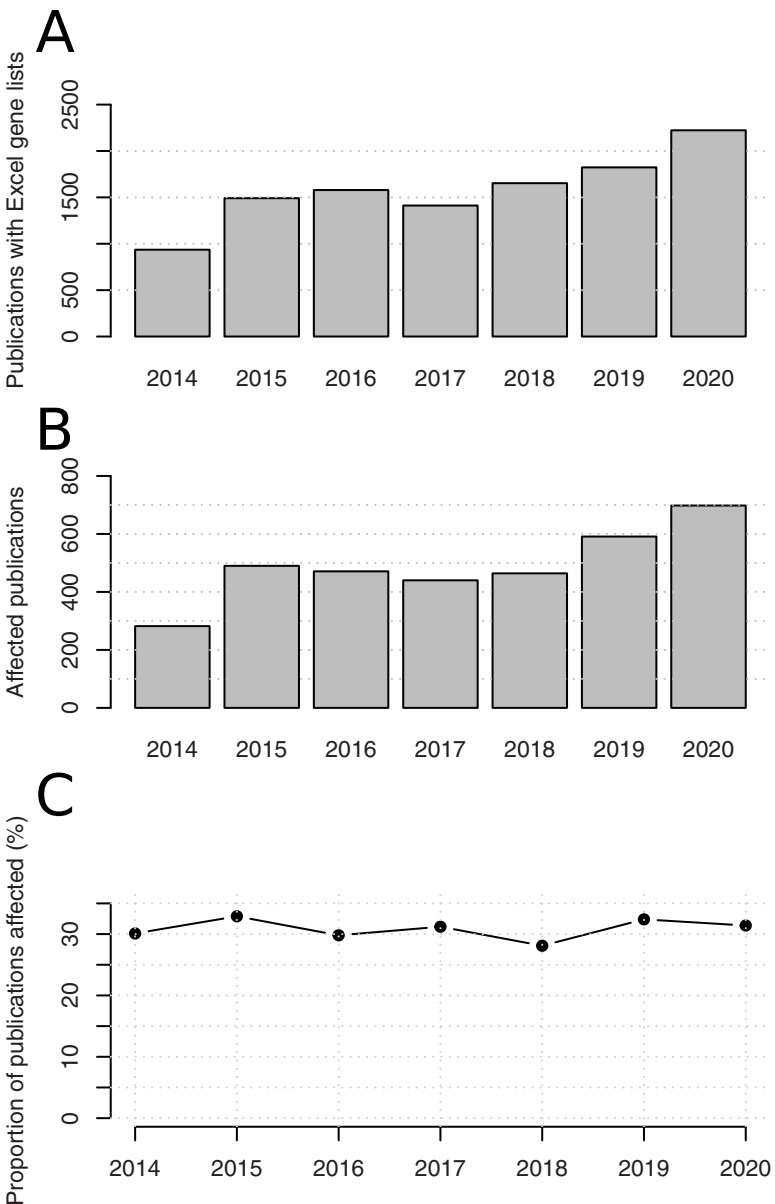

**Fig 1. Prevalence of gene name errors in the period 2014–2020.** (A) Publications with supplementary Excel gene lists. (B) Publications affected by gene name errors. (C) Proportion of affected publications.

(PMC5908809), *TAMM41* was apparently converted to "Jan-41" due to similarity with the month of January in Finnish (tammikuu).

There were also several cases where the dates appeared to be unrelated to Excel date genes. For example, article PMC6330011 S4 Table contained the following: "'Feb-97, Aug-97, Nov-97, Feb-98, Aug-98". Information in other columns of the spreadsheet indicated that these originated from *SEPT*, *MARCH* and *DEC* gene names. Cells containing Aug-97 through Aug-11 corresponded to *SEPT2* to *SEPT14* and *SEP15*. Article PMC5989470 showed evidence that the protein name "jun-1" was converted to "May-31". We posit that this type of error is caused by the spreadsheet evaluating protein names like "jun-1" as the month of June minus 1.

**Table 4. Gene name errors stratified by organism under study.**

| Species | Publications with Excel gene lists | Affected publications | Proportion of publications affected |
|---|---|---|---|
| *M. musculus* | 1577 | 609 | 38.6% |
| *H. sapiens* | 7936 | 2419 | 30.5% |
| *C. elegans* | 124 | 31 | 25.0% |
| *D. melanogaster* | 607 | 142 | 23.4% |
| *S. cerevisiae* | 443 | 93 | 21.0% |
| *R. norvegicus* | 327 | 68 | 20.8% |
| *D. rerio* | 251 | 48 | 19.1% |
| *A. thaliana* | 511 | 76 | 14.9% |
| *G. gallus* | 1827 | 172 | 9.4% |
| *O. sativa* | 10 | 0 | 0.0% |

Other observations were more puzzling. There were two papers where it appears the *P2RY1* gene (Ensembl identifier ENSGALG00000016687) was converted to "7"; possibly a problem in an upstream database. In one sheet (Table S5 of article PMC6506828), the numeric value "3002" was observed in the gene symbol column beside "NM_198411", corresponding to *Inverted Formin 2* (*Inf2*). Perhaps the spreadsheet interpreted "Inf2" as a numerical value.

## Discussion

We hypothesized that after a previous publication in 2016 received substantial attention in technology and social media spheres, that researchers and publishers would be aware of the issue of gene name conversion in spreadsheets and the prevalence of such errors would decline. On the contrary, this work demonstrates that overall there has been no substantial change in the rate of gene name errors in the period 2014 to 2020. Indeed the proportion of articles with Excel gene lists containing gene name errors was significantly higher here as compared to a previous report (30.9% and 19.6% respectively) [2]. This is due to two main contributors. Firstly, the articles here were sampled from PMC as compared to a set of 18 major genomics journals. Secondly this work identified gene names becoming converted to internal date format, which accounts for ~15% of such errors detected here. These numbers correspond to the number of days since 1st January 1900; indeed, this is how spreadsheet software stores date information internally. Gene names can become converted to five-digit numbers by first converting them to dates upon import, followed by changing the cell formatting to "number" or "text", becoming permanent when the spreadsheet is saved.

Another take-away from this study is that articles with supplementary Excel gene lists in highly reputable journals like *Cell*, *Nature* and *Proc Natl Acad Sci USA* more frequently contained gene name errors as compared to their counterparts with lower JIF scores. This may seem counterintuitive, but is consistent with previous analysis [2]. Although it has been suggested that articles in highly prestigious journals are of an inferior methodological quality [6], the simpler explanation is that the number and size of supplementary gene lists accompanying articles is the main contributor to this trend (although we have not examined this hypothesis quantitatively). This is likely a contributing factor to why so many gene name errors were identified in *Nature Communications*. This journal recommends authors provide source data which contain the raw data underlying any graphs and charts, resulting in more data in attached Excel files. Additionally, this is a prolific and fast-growing multidisciplinary journal with 6,448 published articles in 2020 and ~15% year-on-year growth since 2014. Concerningly, the proportion of papers in *Nature Communications* with supplementary Excel gene lists affected by gene name errors also increased in the period 2014–2020 (Fig 3).

**Table 5. Prevalence of gene name errors across journals.** Only journals with ≥50 articles with supplementary Excel gene lists are shown.

| Journal name as it appears in PMC | Number of articles with Excel gene lists | Number of affected articles | Proportion of articles affected (%) |
|---|---|---|---|
| Nat Commun | 920 | 345 | 37.5% |
| PLoS One | 946 | 244 | 25.8% |
| Sci Rep | 767 | 227 | 29.6% |
| BMC Genomics | 660 | 166 | 25.2% |
| PLoS Genet | 448 | 134 | 29.9% |
| Oncotarget | 326 | 107 | 32.8% |
| Front Genet | 313 | 94 | 30.0% |
| eLife | 243 | 89 | 36.6% |
| Proc Natl Acad Sci USA | 155 | 73 | 47.1% |
| Cell Rep | 158 | 71 | 44.9% |
| Genome Biol | 193 | 66 | 34.2% |
| Nature | 118 | 52 | 44.1% |
| Nat Genet | 140 | 48 | 34.3% |
| Genome Med | 137 | 44 | 32.1% |
| PeerJ | 137 | 39 | 28.5% |
| Cell | 74 | 39 | 52.7% |
| Clin Epigenetics | 109 | 38 | 34.9% |
| Nucleic Acids Res | 120 | 36 | 30.0% |
| BMC Med Genomics | 117 | 31 | 26.5% |
| Front Oncol | 85 | 31 | 36.5% |
| Transl Psychiatry | 73 | 29 | 39.7% |
| BMC Cancer | 105 | 28 | 26.7% |
| PLoS Pathog | 80 | 27 | 33.8% |
| Commun Biol | 74 | 27 | 36.5% |
| PLoS Biol | 66 | 26 | 39.4% |
| Aging | 56 | 26 | 46.4% |
| EBioMedicine | 51 | 26 | 51.0% |
| Epigenetics Chromatin | 64 | 25 | 39.1% |
| PLoS Comput Biol | 97 | 24 | 24.7% |
| Oncogene | 53 | 22 | 41.5% |
| iScience | 58 | 20 | 34.5% |
| Sci Adv | 56 | 20 | 35.7% |
| BMC Bioinformatics | 77 | 19 | 24.7% |
| G3 | 74 | 15 | 20.3% |
| Hum Mol Genet | 53 | 15 | 28.3% |
| BMC Plant Biol | 52 | 6 | 11.5% |
| Front Plant Sci | 75 | 5 | 6.7% |

There are limitations to this study that need to be pointed out. For convenience, we only screened open access articles in PMC and so this might not be representative of the work in paywalled articles. Moreover, we screened a subset of PMC articles that contained the keyword "genom*" in the abstract or title. Out of 3,291,704 articles in PMC published in the period 2014–2020, we included only 116,139 (~5.0%). There are likely many gene name errors outside of this sample of articles and there is a chance that such errors appear at varying rates in the articles not analysed here. The updated screening software yielded a slightly higher fraction of false positives but was circumvented by systematically opening each file manually for

## Articles with supplementary Excel gene lists

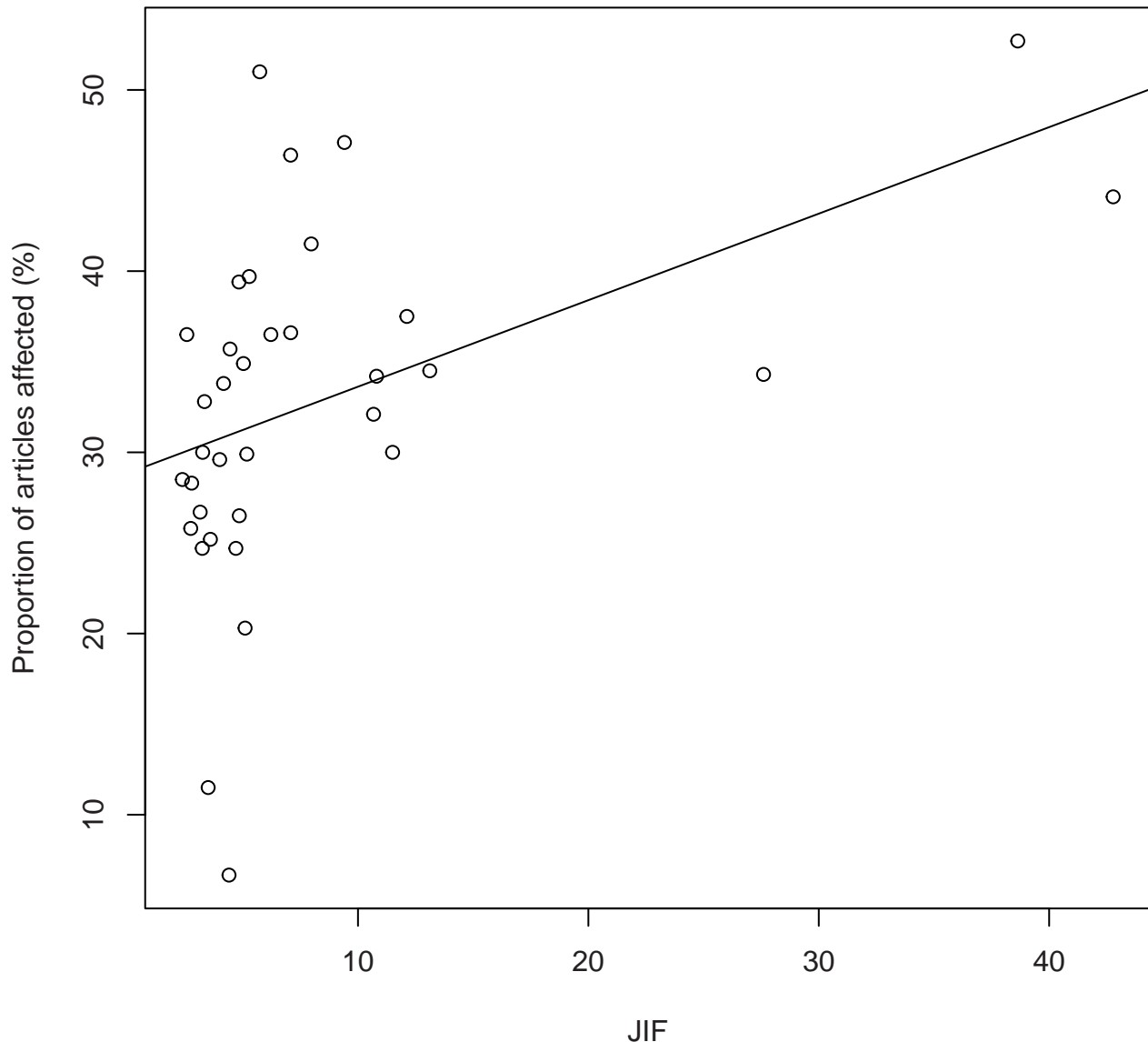

**Fig 2. A scatterplot of JIF and proportion of articles with supplementary Excel gene lists affected by gene name errors.**

verification. Our script only identified vertical gene lists, so there were likely some in the horizontal orientation that were missed.

There has been a great deal of discussion around who is responsible for the persistence in gene name errors over time. The software developers surely must take some blame because these conversions occur without any user notifications, and the date conversion feature is not one that can be disabled. In their defence, we must understand that Excel and other spreadsheet software were designed only for lightweight data entry and calculation, not for analysis of data containing many thousands of rows. Reviewers are doing their best with limited time but can do better with regards to quality checking supplementary files. Journal editors have yet

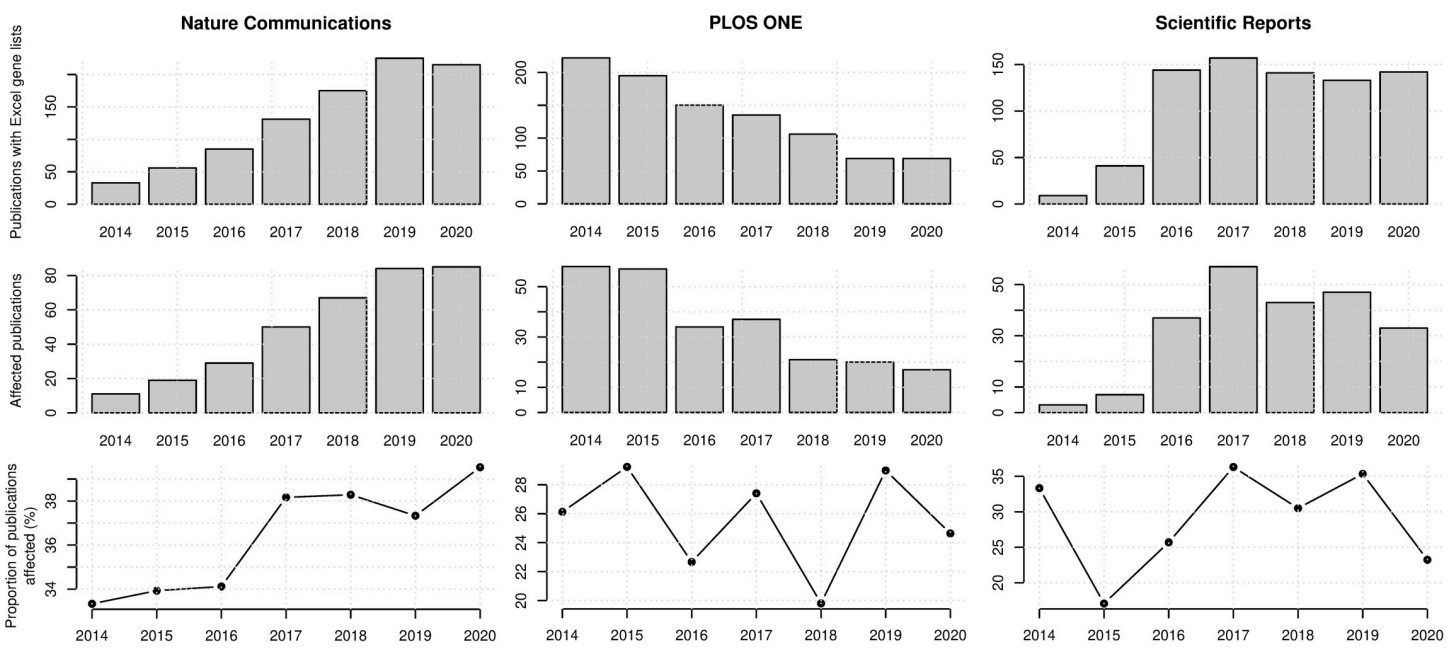

**Fig 3. Gene name errors in supplementary files for three dominant journals in the period 2014–2020.**

to put in place systems to identify gene name errors before they are published. Surely some blame rests on the researchers who inadvertently make these mistakes. In particular, senior authors need to take leadership in picking up such errors when they arise, but more importantly, they need to provide training opportunities and promote a culture of reproducibility in the groups they lead. Academic faculty need to ensure that biology graduates are trained in contemporary skills to conduct data-driven research that goes beyond appropriate use of spreadsheets. This needs to include competence in scripted computer languages, statistical analysis and computational reproducibility [5].

From the researcher's perspective, there are several practical ways that such errors can be avoided (Box 1).

The HGNC has taken the initiative to change the most susceptible gene names, but this will not entirely solve the problem. There are a number of gene names that could be converted if the user computer is set up to use a non-English language. While human, mouse, and rat gene names have been changed, such changes are yet to take place for other species such as *D. rerio*, *C. elegans*, *D. melanogaster* and *A. thaliana* (See S1 Table). Open-source tools are being developed to circumvent these errors. Truke is a web service that identifies and corrects corrupted gene names in affected files [7], while EscapeExcel is a tool designed to prevent gene name conversions from happening by protecting strings before import [8]. HGNChelper is an R package that recognises and fixes human gene symbols converted to dates [9]. It appears that these developments are not having a major impact yet because gene name errors continue to grow year-on-year and the proportion of affected articles has remained stable since 2014 (Table 3 and Fig 1).

It has been argued that gene name errors are of little consequence to the conclusions of a scientific publication [10], however our view is that it is a symptom of a larger problem—that overreliance on spreadsheets leads to errors occurring silently in large data files and that such errors are exceedingly difficult for researchers, reviewers, and editorial staff to identify. Previous spreadsheet research in the business setting indicates that errors exist in 0.9% to 1.9% of

## Box 1. Tips to avoid gene name errors

- Scripted analyses are preferred over spreadsheets. Gene name to date conversion is a bug specific to spreadsheets and doesn't occur in scripted computer languages like Python or R. In addition, analyses conducted with Python and R notebooks (eg: Jupyter or Rmarkdown) capture computational methods and results in a stepwise fashion meaning these workflows can be more readily audited. These notebooks can therefore achieve a higher level of computational reproducibility than spreadsheets. Although this requires a big investment in learning a computer language, this investment pays off in the longer term.

- If a spreadsheet must be used, then LibreOffice is recommended because it will avoid such errors from occurring. This will not remedy other error types.

- If using Excel is unavoidable, then take great care importing the data. If opening a TSV or CSV file, use the data import wizard to ensure that each column of data is formatted appropriately. For example, columns containing gene names should be formatted as "free text", genomic coordinates formatted as "integers" and gene expression measurements as "numeric".

- Instead of spreadsheets, share genomic data as "flat text" files. These typically have the suffixes "csv", "tsv" or "txt". These are native formats for computer languages and suitable for long term data archiving. Excel formats such as "xls" or "xlsx" are proprietary and future development is decided by Microsoft.

- If it is unavoidable to use a spreadsheet with genomic data, verify that gene names are intact. To do this, sort columns containing gene names in ascending order. This will bring dates and numbers to the top of the column so it is obvious whether any gene symbols have been converted. Alternatively, use the Truke web tool to identify such errors (http://maplab.imppc.org/truke/).

- Assume that there are Excel date gene names in your organism of interest. Although human and mouse SEPT and MARCH gene names have been changed to avoid such errors, there are many taxa across Eukarya that are yet to see similar changes. Excel gene names may also be prevalent in Prokarya.

formula cells and from a sample of 50 spreadsheets, only seven were error-free [11]. In the healthcare sector, an analysis of data entry errors into a clinical pathology spreadsheet found errors in 0.5% to 6.4% of cells [12], while a systematic analysis of spreadsheet errors in a hospital setting found critical errors in 11 of 12 spreadsheets analyzed [13]. In the biomedical research setting we know that spreadsheet errors can occur and impact downstream work involving clinical drug trials [14]. Despite this potential risk, there has yet to be a systematic assessment of the full taxonomy of spreadsheet errors in biomedical research, so we don't know how frequently they occur.

It must be noted that a blanket ban on spreadsheets as supplementary files is unlikely to mitigate gene name errors entirely, as many researchers might simply export their working

spreadsheets to flat text files (errors included). Rather, raising standards around computer code sharing, code review, and reproducibility measures is more likely to deliver lasting improvements in the quality of published research.

In summary, this work demonstrates that gene name errors in supplementary data files of research articles are more frequent than previously appreciated and are not declining over time. Eliminating gene name errors will require major changes to researcher practices which are unlikely to happen in the near term. To monitor gene name errors in PMC we have set up an automated reporting system that will be updated monthly (URL: http://ziemann-lab.net/public/gene_name_errors/).

## Methods

### Characterising spreadsheet software behaviours

We tested the default behaviour of four different spreadsheet programs (Microsoft Excel 365 MSO version, Google Sheets (accessed 4th June 2021), LibreOffice v6.4.6.2, and Gnumeric v1.12.46) by entering the list of strings shown in Box 2. These data were entered into spreadsheets by (i) opening directly from a text file with csv or tsv suffix, (ii) typing directly into cells, and (iii) pasting from a separate text file. We then observed and recorded the propensity of these programs to perform date conversion of the gene symbols.

---

### Box 2. Strings used to test date conversion of spreadsheet programs

1/1/2001

2/3/2001

5/4/2008

SEPT7

DEC1

OCT4

MARCH3

5/1/2010

TP53

NCF1

Inf2

---

### Screening gene names that get converted to dates

All eukaryotic gene annotation files were downloaded from Ensembl (Vertebrates v102, Metazoa v49, Plants v49, Fungi v49 and Protists v49). Gene names were extracted from the GTF files and imported into Excel together with the taxa name (species/strain). The gene name column was sorted to bring cells containing dates to the top of the sheet, where we counted the number of date conversions per taxon.

### Searching PMC

PMC (URL: https://www.ncbi.nlm.nih.gov/pmc/) was our starting point for shortlisting open-access publications to screen. We did not screen every publication in PMC because most do not include genomic data. By searching for publications with the keyword "genom*" in the title or abstract, we were able to reduce the number of articles screened by ~95%. For example, in the year 2015, there were 405,251 articles published, but only 21,213 had the keyword "genom*" in the title or abstract. We used this approach to create lists of PMC identifiers by year for the period 2014 to 2020.

### Updated software for scanning for gene name errors

A shell script was used to perform the following. Each PMC publication in the shortlist was downloaded as a HTML file. Links to files with.xls or.xlsx suffixes in the HTML were extracted, these were assumed to be supplementary Excel files. Each Excel file was downloaded, and file metadata was scanned to confirm it is an Excel file and not simply a tabular text file with an incorrect suffix. True Excel files were extracted with an R script (R v4.0.0) using the readxl package v1.3.1 (https://CRAN.R-project.org/package=readxl) into tabular files. Other text-based files with xls or.xlsx suffices were processed with ssconvert v1.12.46 to tabular files. As per a previous study [2], these tabular data underwent screening for columns that contained gene symbols. Those columns with five or more gene symbols were considered to be gene lists and underwent screening for erroneous conversions, such as date formats and scientific numbers. The main difference being that this script also recognises five-digit numbers (internal date format). Analysis logs were processed and brought together with the corresponding journal name to yield a list of supplementary files suspected to contain a gene name error.

### Verification and data visualisation

Each of these suspect files were downloaded and opened with either Excel or LibreOffice Calc to confirm the presence of gene name errors. To do this, columns appearing to contain gene names were sorted such that numeric values (dates) were brought to the top of the sheet. Summary data were loaded into R v4.1.0 for analysis and visualisation. The two-sided Pearson and Spearman correlation tests were executed in R.

## Supporting information

**S1 Table. Ensembl eukaryotic gene names converted to dates by Excel.**
(XLSX)

**S2 Table. Articles and supplementary Excel files affected by gene name errors.**
(XLSX)

**S3 Table. Excel files suspected to contain gene name errors by the screening software but turned out to be false positives.**
(XLSX)

**S4 Table. Classification of gene name error types in a sample of 2160 affected spreadsheets.**
(XLSX)

## Acknowledgments

We thank Dr Nicholas C. Wong from Monash University for comments on the manuscript.

## Author Contributions

**Conceptualization:** Mark Ziemann.

**Data curation:** Mandhri Abeysooriya, Mark Ziemann.

**Formal analysis:** Mandhri Abeysooriya, Mark Ziemann.

**Investigation:** Mandhri Abeysooriya, Megan Soria, Mary Sravya Kasu, Mark Ziemann.

**Methodology:** Mandhri Abeysooriya, Mark Ziemann.

**Project administration:** Mark Ziemann.

**Resources:** Mark Ziemann.

**Software:** Mandhri Abeysooriya, Mark Ziemann.

**Supervision:** Mark Ziemann.

**Validation:** Mandhri Abeysooriya, Megan Soria, Mary Sravya Kasu, Mark Ziemann.

**Visualization:** Mandhri Abeysooriya, Mark Ziemann.

**Writing – original draft:** Mandhri Abeysooriya, Mark Ziemann.

**Writing – review & editing:** Mandhri Abeysooriya, Megan Soria, Mary Sravya Kasu, Mark Ziemann.

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
