## [Decision Letter · Decision Letter 0]

2 Jun 2021

Dear Dr Ziemann,

Thank you very much for submitting your manuscript "Gene name errors: lessons not learned" for consideration at PLOS Computational Biology. As with all papers reviewed by the journal, your manuscript was reviewed by members of the editorial board and by several independent reviewers. The reviewers appreciated the attention to an important topic. Based on the reviews, we are likely to accept this manuscript for publication, providing that you modify the manuscript according to the review recommendations.

Sincerely,

Christos A. Ouzounis

Associate Editor

PLOS Computational Biology

Ilya Ioshikhes

Deputy Editor

PLOS Computational Biology

[LINK]

Reviewer's Responses to Questions

**Comments to the Authors:**

Reviewer #1: The authors have written a clearly readable paper with all relevant data and code made available.

It is beneficial to highlight how incorrect gene names continue to be published and how this can be an issue as they are difficult to check for.

It is unclear how useful the comparison between different publications is. It is unclear how many other users will see value in the provided software. It is also unclear whether it was necessary to spend time focusing on which organisms have gene names at risk, and the text devoted to gene name errors by year. Calling a correlation coefficient of 0.589 a strong association is somewhat optimistic.

This paper could be strengthened by offering more solutions to this problem, for example by emphasising using .csv rather than .txt format, and alternatives to spreadsheet software, such as RStudio.

A couple of minor points, it is possible to paste directly into Google Sheets without errors occurring. There appears to be an error with the automated reporting system, with four identical reports currently shown.

Reviewer #2: I applaud the hard work of the authors. Opening several thousand file by hand is a heroic task that nobody envies them for. My hat is off!

I liked the analysis and the writing very much. I have only four minor points, the last one is optional:

1. Ordering table 4 by highest to lowest „Proportion of publications affected” makes grasping the results much easier.

2. Show graph for the correlation between JIF and error rate and report R2. Such a graph is useful for visualizing the negative correlation of journal rank with reliability, consistent with similar graphs from other publications (see next point).

3. In the discussion (p14), the authors mention that the correlation of error rate with journal prestige may seem counterintuitive, but it is not only the 2016 analysis that their results are consistent with. See, e.g., https://doi.org/10.3389/fnhum.2018.00037 for a list of many other analyses that all suggest that work published in prestigious journals is more error prone than that of other publication venues. I fact, it would have been counter-intuitive had the authors found less errors in highly prestigious journals: the evidence predicts more errors in more prestigious journals.

4. The authors may consider inserting a few sentences into their discussion about a solution to this (and many other, related) problems with the data and code underlying our narratives: an infrastructure solution that integrates research data and code in a way that makes supplemental files unnecessary. Such a solution has been technically feasible for some years now and many people have suggested to use such solutions, e.g.:

https://journals.plos.org/plosbiology/article?id=10.1371/journal.pbio.3000117

http://www.theguardian.com/science/head-quarters/2015/may/12/will-traditional-science-journals-disappear

https://neuroneurotic.net/2015/07/17/revolutionise-the-publication-process/

https://micahallen.org/2015/03/20/short-post-my-science-fiction-vision-of-how-science-could-work-in-the-future/

Björn Brembs

**Have the authors made all data and (if applicable) computational code underlying the findings in their manuscript fully available?**

Reviewer #1: Yes

Reviewer #2: Yes

PLOS authors have the option to publish the peer review history of their article (what does this mean?). If published, this will include your full peer review and any attached files.

Reviewer #1: No

Reviewer #2: **Yes: **Björn Brembs

Figure Files:

Data Requirements:

Reproducibility:

References:

---

## [Decision Letter · Decision Letter 1]

1 Jul 2021

Dear Dr Ziemann,

We are pleased to inform you that your manuscript 'Gene name errors: lessons not learned' has been provisionally accepted for publication in PLOS Computational Biology.

Best regards,

Christos A. Ouzounis

Associate Editor

PLOS Computational Biology

Ilya Ioshikhes

Deputy Editor

PLOS Computational Biology

Reviewer's Responses to Questions

**Comments to the Authors:**

Reviewer #1: I thank the Authors for addressing all of the comments. I am pleased with the additions, particularly Box 1. This article helps continue to strengthen the case for reproducible and transparent science, thank you.

Reviewer #2: The reivewers have addressed all my concerns adequately.

**Have the authors made all data and (if applicable) computational code underlying the findings in their manuscript fully available?**

Reviewer #1: Yes

Reviewer #2: Yes

PLOS authors have the option to publish the peer review history of their article (what does this mean?). If published, this will include your full peer review and any attached files.

Reviewer #1: No

Reviewer #2: **Yes: **Björn Brembs

---

## [Editor Report · Acceptance letter]

22 Jul 2021

PCOMPBIOL-D-21-00825R1 

Gene name errors: lessons not learned

Dear Dr Ziemann,

I am pleased to inform you that your manuscript has been formally accepted for publication in PLOS Computational Biology. Your manuscript is now with our production department and you will be notified of the publication date in due course.

With kind regards,

Zsofi Zombor
